# Sulfated Glucan from the Green Seaweed *Caulerpa sertularioides* Inhibits Adipogenesis through Suppression of Adipogenic and Lipogenic Key Factors

**DOI:** 10.3390/md20080470

**Published:** 2022-07-23

**Authors:** Gildacio Pereira Chaves Filho, Lucas Alighieri Neves Costa Batista, Silvia Regina Batistuzzo de Medeiros, Hugo Alexandre Oliveira Rocha, Susana Margarida Gomes Moreira

**Affiliations:** 1Laboratory of Molecular and Genomic Biology, Department of Biology and Genetics, Center of Biosciences, Federal University of Rio Grande do Norte, Natal 59072-900, RN, Brazil; gildacio_@hotmail.com (G.P.C.F.); silvia.batistuzzo@ufrn.br (S.R.B.d.M.); 2The Doctoral Program in Biotechnology—Northeast Biotechnology Network (RENORBIO), Center of Biosciences, Federal University of Rio Grande do Norte, Natal 59072-900, RN, Brazil; hugo.rocha@ufrn.br; 3Laboratory of Biotechnology of Natural Polymers, Department of Biochemistry, Center of Biosciences, Federal University of Rio Grande do Norte, Natal 59072-900, RN, Brazil; lucasalighieri@gmail.com

**Keywords:** marine algae, anti-obesity, polysaccharides

## Abstract

Sulfated polysaccharides (SPS) from seaweeds have great biochemical and biotechnological potential. This study aimed to investigate the effect of SPS isolated from the seaweed *Caulerpa sertularioides* on adipogenic differentiation as a possible alternative treatment for obesity. The SPS-rich extract from the seaweed *C. sertularioides* was fractioned into three SPS-rich fractions (F0.5; F0.9; and F1.8) chemically characterized. Among these four samples, only F0.9 showed a significant inhibitory effect on adipogenesis of 3T3-L1 preadipocytes. Ten SPS-rich fractions were isolated from F0.9 through ion-exchange chromatography. However, only the fraction (CS0.2) containing a sulfated glucan was able to inhibit adipogenesis. CS0.2 reduces lipid accumulation and inhibits the expression of key adipogenic (PPARγ, C/EBPβ, and C/EBPα) and lipogenic markers (SREBP-1c, Fabp4, and CD36). The data points to the potential of sulfated glucan from *C. sertularioides* for the development of functional approaches in obesity management.

## 1. Introduction

Obesity is a worldwide public health problem correlated with the development of other serious diseases, such as hypertension, cardiovascular disease, and type 2 diabetes mellitus, among others [1,2], and whose treatment and prevention is a global public health concern [1].

Synthetic anti-obesity drugs are important for the treatment of weight loss but also for the amelioration of the aforementioned diseases [1,3,4]. Yet, the use of these drugs is associated with serious negative side effects, with gastrointestinal disturbances, oily stools, incontinence, and cases of pancreatitis being described [5]. Therefore, significant attention has been paid to natural compounds to develop new approaches to preventing and treating obesity and its related diseases [6,7,8].

The development of obesity results from a complex multi-step process that involves the differentiation of preadipocytes in adipocytes (adipogenesis), resulting in the hypertrophy and proliferation of adipose tissue [9]. The transcriptional factors cytosine-cytosine-adenosine-adenosine-thymidine (CCAAT)/enhancer-binding protein (C/EBP-α/β/γ subunits) and peroxisome proliferator-activated receptor gamma (PPAR-γ subunit), play critical roles in this process [10]. Thereafter, sterol regulatory element-binding protein-1c (SREBP-1c) promotes the expression of PPARγ, which is also induced by C/EBP factors. Then, the expression of lipogenesis-associated genes, such as fatty acid-binding protein (Fabp4) and the cluster of differentiation 36 (CD36), are finally initiated [11]. Therefore, has been suggested that an approach capable of decreasing the expression of these main transcription factors may be a potential intervention in the prevention and treatment of obesity [9]

Natural compounds are evaluated for anti-adipogenic activity using in vitro models and methods that can in vivo control obesity [6,10,12,13,14]. Inclusive, it has been shown that natural products can minimize body weight and fasting blood glucose levels and improve insulin resistance through clinical studies [10,15].

Seaweed products have aroused great interest for applications in nutraceutical and pharmaceutical industries due to their bioactive potential and reported health benefits associated with their consumption. Additionally, the results of in vitro and in vivo studies show that seaweed extracts reduce the expression of adipogenesis controlling factors [6,16]. Among the different seaweed constituents, the sulfated polysaccharides (SPS) [17,18], terpenoids [19], and polyphenols [20] showed important biological activities.

Tropical seaweeds also synthesize SPS that interfere with lipid metabolism. For instance, a sulfated agaran produced by the *Gracilaria birdiae*, a red seaweed, showed anti-adipogenic activity in vitro and decreased weight gain in mice when they received food supplemented with agaran [21]. Fucoidan, an SPS synthesized by brown seaweeds, also showed the capacity to suppress adipogenesis of 3T3-L1 [22,23] and manage diabetes by stimulating glucose uptake and lowering basal lipolysis [24]. In addition, Oliveira et al. (2018) fractionated the antiadipogenic fucoidan from *Fucus vesiculosus*. They obtained four fractions, and two of those fractions showed anti-adipogenic action, while the other two had pro-adipogenic action. This indicates that SPS-rich extracts may contain SPS with anti- or pro-adipogenic action and, therefore, it is necessary to evaluate the action of isolated SPS.

Concerning green seaweed, in an initial screening, we reported the first result showing the anti-adipogenic effect of the SPS-rich sample from *Caulerpa prolifera* [25]. However, there are several species of the genus Caulerpa in the Brazilian Northeast, and we evaluated the anti-adipogenic action of SPS-rich extracts from several Caulerpa species (unpublished data). Among them, the SPS-rich extract from *C. sertularioides* stood out. Therefore, using sequential butanone precipitation, three fractions of SPS-rich extract from *C. sertularioides* were obtained, and their in vitro cytotoxicity, adipogenesis, and lipogenesis activities were evaluated using 3T3-L1 cells. In addition, the polysaccharides from that fraction with the best anti-adipogenic action were purified by ion-exchange chromatography, and their effect on anti-adipogenic differentiation was evaluated.

## 2. Results

### 2.1. Extract Characterization and Screening of Anti-Adipogenic Activity

Results from the chemical composition of crude extract and butanone fractions (F0.5, F0.9, and F1.8) and the yield (%) of each fraction obtained are presented in Table 1. The yield of each fraction was calculated based on the amount of crude extract.

The effects of SPS-containing samples on adipogenic cell differentiation will be further detailed in Section 2.2. The samples used in the initial adipogenic potential screening did not present cytotoxicity in 3T3-L1 cells, as MTT results show (Figure 1A. Only the F0.9 fraction (composed of 59.1% of total sugars and 11.2% of sulfate) showed a significant inhibitory effect on adipogenesis (Figure 1B). Therefore, the F0.9 sample was further fractionated, by ion-exchange chromatography, resulting in 10 fractions, whose chemical composition is in Table 2.

All new fractions present sugar and sulfate in their composition, without protein traces, and among all samples, CS0.2 showed a higher yield. Concerning monosaccharide characterization, CS1.0 did not have its composition characterized due to a lack of sufficient material. Furthermore, CS0.1 did not have its monosaccharide composition determined because it corresponds to the residual material of F0.9 that was not trapped in the column. Results showed that glucose was the monosaccharide present in all fractions (CS0.2 to CS0.9), being predominant in CS0.2 and CS0.7. Except for the CS0.2 fraction, all samples have xylanose in different proportions, which is the predominant monosaccharide in the CS0.9 fraction. Mannose is also present in nearly all samples (except for CS0.3 and CS0.4), but in sample CS0.5, it is the predominant monosaccharide. Galactose is the predominant monosaccharide in CS0.8. The CS0.3 also has large amounts of galactose, but the predominant monosaccharide is rhamnose, which is found in a 1:2 ratio (Table 2).

The FITC spectra of the 9 fractions are shown in Figure 2. As can be seen, they are very similar and show characteristic bands of sulfated polysaccharides obtained from algae of the genus Caulerpa. A broadband (3430–3480 cm^−1^), a narrower band (2920–2930 cm^−1^), and two moderately widened bands (1640–1650 cm^−1^ and 1110–1010 cm^−1^), which correspond to the stretch vibration of the O-H, C-H, C=O, and C-O-C bonds (glycosidic bond), respectively [26]. In the region between 1200–1300 cm^−1,^ there is an absorption band designated as asymmetrical stretch S=O, which is characteristic of the sulfate group [27].

The results showed that only the CS0.2 presented a significant inhibitory effect on 3T3-L1 cells adipogenesis (Figure 1C), which was therefore selected to proceed with this study. This sample has approximately 63.6% total sugars and 2.8% sulfate in its composition (Table 2), the sample with the lowest sulfate content.

### 2.2. Effects of SPS from C. sertularioides on Adipocyte Differentiation

After initial cytotoxicity screening, only F0.9 at 200 μg mL^−1^ suppressed fat accumulation in about 45.97%, suggesting inhibition of adipogenesis (Figure 1).

The nine fractions obtained from the F09 sample were used in the second screening of adipogenic activity. However, only the CS0.2 sample showed a significant inhibitory effect on 3T3-L1 cell differentiation (Figure 1C). No cytotoxicity was detected for the concentrations tested (25, 50, 100, and 200 μg mL^−1^) (Figure 3), and the minimum concentration needed to obtain an anti-adipogenic effect was 50 μg mL^−1^ of SPS (Figure 4). This concentration lowered fat accumulation of 3T3-L1 cells by about 53%; therefore, it was selected to proceed with gene expression analysis.

### 2.3. Effect of SPS from C. sertularioides on the Expression of Adipogenic and Lipogenic Markers

To investigate how SPS from *C. sertularioides* inhibited adipogenesis in 3T3-L1 cells, we first measured the gene expression of adipogenic markers. As shown in Figure 5, RT-PCR results revealed that PPARγ, C/EBPβ, and C/EBPα mRNA levels were reduced by about 6.0, 3.0, and 3.6-fold, respectively, in cells treated with CS0.2 sample at 50 μg mL^−1^.

To confirm whether SPS from *C. sertularioides* inhibits lipogenesis, we determined the expression level of lipogenic transcription factors. RT-PCR results revealed that the expression of SREBP-1c, Fabp4, and CD36 was also significantly reduced by about 3.0, 5.4, and 9.0-fold, respectively, following treatment with CS0.2 sample at 50 μg mL^−1^ (Figure 5).

## 3. Discussion

An increase in the number and size of adipocytes characterizes obesity, which is a complex condition that results from a combination of energy balance and individual factors [9].

Regarding drug management, the most commonly used drugs approved by the FDA are orlistat [2], liraglutide [28], and sibutramine [29], an irreversible pancreatic lipase inhibitor, a GLP-1 receptor agonist that suppresses appetite, and an inhibitor of the reuptake of serotonin and norepinephrine, respectively. However, there is a growing need for developing anti-obesity treatments based on natural products with low side effects and dependency rates. Biomolecules derived from natural products have the potential to be used as models for the design and production of analogs, resulting in more efficient and less toxic therapies. Based on this premise, we here evaluate the effect of SPS from *C. sertularioides* on adipocyte differentiation through experiments in 3T3-L1 cells.

First, the chemical composition of the crude extract, the 3 fractions, and the 10 fractions were characterized. The results (Table 1 and Table 2) showed that SPS-containing samples are composed of sugar and sulfate. The FTIR results confirmed the presence of polysaccharides and sulfate in the fractions (Figure 2), and the differences between the SPS samples spectra showed less sulfate ratio in CS0.2-CS0.5 and higher sulfate ratio in CS0.8-CS0.9, probably due to the increased NaCl molarity used to elute them.

Once the CS0.2 fraction was not cytotoxic (Figure 3), its anti-adipogenic effect was evaluated. The Oil Red O staining results showed that the CS0.2 fraction significantly reduced the fat accumulation in 3T3-L1 preadipocytes (Figure 4). Moreover, the highest effect on fat accumulation reduction was obtained using a lower concentration compared to F0.9 extract. Moreover, this is the lower concentration described compared with other studies using the same cellular model. For example, in a previous study, we showed that SPS from *C. prolifera* (a green seaweed), at 100 and 200 μg mL^−1^, decreased fat accumulation by about 57.6% [25]. It was also shown that 100 and 200 μg mL^−1^ of fucoidan from *F. vesiculosus* (Sigma-Aldrich) decreased fat accumulation by about 32.8 and 39.7%, respectively, by Oil Red O staining [22].

The adipogenic process involves a regulated and sequential activation of transcription factors C/EBPα, C/EBPβ, and PPARγ [30], which were evaluated in this study to confirm the anti-adipogenic effect of SPS from *C. sertularioides* in 3T3-L1. Our results showed that the CS0.2 significantly decreased the expression of C/EBPα, C/EBPβ, and PPARγ mRNA levels (Figure 5).

Lipogenesis is an inherent process of adipocytes and includes the synthesis and storage of fatty acids and triglycerides, which increase adipose tissue mass [30]. SREBP1c, Fabp4, and CD36 have critical roles in lipogenesis: for example, the activation of SREBP-1c induces the expression of several lipogenic enzymes, such as Fabp4, which in turn is a downstream target of PPARγ and C/EBPα. CD36 is a marker of human adipocyte progenitors with triglyceride storage potential [9]. In this study, cells treated with SPS from *C. sertularioides* showed reduced expression levels of the key lipogenic markers. This result is consistent with the expected downregulation of PPARγ (Figure 5), which is the master adipocyte regulator [30].

Thus, we hypothesize that SPS from *C. sertularioides* regulates adipocyte differentiation through inhibition of early adipogenic factors, such as C/EBPβ, and also represses the expression of C/EBPα and PPARγ, which are adipogenic transcriptional factors [8,31]. Finally, these processes lead to the downregulation of lipogenic genes SREBP1c, Fabp4, and CD36.

Recently, it has been shown that natural anti-obesity agents could modulate transcription factors in managing obesity and its related diseases. Jee and colleagues showed that a polyphenol (*Polygala japonica* Houtt.) inhibits adipocyte 3T3-L1 cells differentiation by suppressing adipogenic and lipogenic factors [13]. Park et al. (2020) also showed the anti-adipogenic effect of ethanolic extract of *Ramulus mori* on 3T3-L1 adipocytes with down-regulation of mRNA of adipogenic and lipogenic markers. They also showed an anti-obesity effect in the obese mice model [14].

Regarding the anti-adipogenic activity of SPS from seaweed, we showed in a previous study that the SPS-rich fraction from *C. prolifera* suppresses all mentioned adipogenesis and lipogenesis markers analyzed here [25]. Furthermore, Kim and colleagues showed that fucoidan from the brown alga *Undaria pinnatifida* suppresses adipocyte differentiation by downregulating adipocyte markers, such as PPARγ and C/EBPα, and then inhibiting adipocyte differentiation through the mitogen-activated protein kinase (MAPK) pathway in 3T3-L1 [22].

Oliveira et al. (2018) described an interesting result. They showed that fractions prepared from commercial fucoidan (*F. vesiculosus*) alter the cell phenotype differently. The 2.0-fraction (20.40% sulfate) reduced-fat accumulation by about 50%, whereas the 0.5-fraction (12.70% sulfate) increased fat accumulation by approximately 80%, relative to the control group [23].

Although fucoidan fractions with different sulfate content have different adipogenic activities, we did not find in the literature a consistent correlation between the sulfate content in the sample and its biological activity. Oliveira et al. (2018) showed that the more sulfated sample obtained from fucoidan had an anti-adipogenic effect and the less sulfated sample had a pro-adipogenic effect. The relation between sulfate content and the adipogenic effect observed in our study is inverse, the more sulfated samples had pro-adipogenic activity, and the less sulfated samples (F0.9 and CS0.2) had anti-adipogenic activity. Furthermore, in previous work, we showed a positive correlation between sulfate content and osteogenic activity of SPS-containing samples from *C. prolifera* [32].

Altogether, the results support the hypothesis that different SPS in a crude extract are a mixture of molecules with different compositions and structures, reinforcing the idea that each SPS has a unique biological activity. Further studies are needed to fully clarify the anti-adipogenesis effect of this sulfated glucan, fully structurally characterize these molecules, and propose hypotheses to correlate their chemical structure and biological activity. However, the present data confirmed that the cell treatment with SPS from *C. certularioides* inhibited the expression of adipogenic and lipogenic markers without a cytotoxicity effect. In addition to their biological effects, the water solubility of these compounds and the worldwide distribution of *C. sertularioides* make these SPS promisors candidates for nutraceutical foods development, with the potential to bring additional health benefits over their nutritional value. Therefore, these SPS are promising natural products for the development of new strategies for the prevention and treatment of obesity and its related diseases.

## 4. Materials and Methods

### 4.1. Extraction and Preparation of SPS-Containing Samples

Specimens of *C. sertularioides* (S.G.Gmelin) were collected with authorization of the Brazilian National System for the Management of Genetic Heritage and Associated Traditional Knowledge (SisGen) n° A0D4240, along the coast of Natal, RN, Brazil. The morphology was used to identify the seaweed specimen, as previously described [33].

Once identified, the specimens were submitted to SPS extraction according to the previously described protocol [25]. Briefly, the seaweeds were dried, powdered, and processed for pigments, lipids, and protein removal. Butanone precipitation was used to obtain 3 fractions containing the SPS from the crude extract. The fractions were named according to the butanone volume used in their preparation: F0.5, F0.9, and F1.8. All fractions were screened for anti-adipogenic effect, and the polysaccharides from that fraction with the best anti-adipogenic action were purified by ion-exchange chromatography.

#### Ion Exchange Chromatography

A column (26 mm inner diameter and 30 cm length) was packed with 150 mL of the weak anionic exchange resin TSKgel DEAE-5PW (20 µm), factored by Tosoh Bioscience LCC. The sample (20 mg mL^−1^ in 0.05 sodium acetate pH 5.00) passed through the column three times to ensure its full complexation on the resin. Then, the column was eluted with 500 mL of sodium acetate solution (0.05 mM and pH 5.00) to remove unbound compounds.

Different concentrations of NaCl (0.1 M; 0.2 M; 0.3 M; 0.4 M; 0.5 M; 0.6 M; 0.7 M; 0.8 M; 0.9 M; and 1.0 M), were used to elute the samples. Excess salt in the eluates was removed by dialysis (MWCO < 6 kDa), and the eluates were lyophilized as described earlier [34]. The new fractions were named according with the concentrations of NaCl used in their preparation: CS0.1, CS0.2, CS0.3, CS0.4, CS0.5, CS0.6, CS0.7, CS0.8, CS0.9. CS1.0. Figure 6 outlines the methodology used for extract preparation.

### 4.2. Analyses of Chemical Characterization of SPS-Containing Samples

#### 4.2.1. Chemical Characterization

To determine the monosaccharide composition, the polysaccharides from *C. sertularioides* were hydrolyzed with 2 M HCl for 2 h at 100 °C and processed as described previously [25]. Once the pH of the resulting material was neutralized, dried, and resuspended in water [35], the monosaccharide composition was determined using a LaChrom Elite HPLC system (VWRHitachi-Chiyoda, Tokio, Japan) with a refractive index detector (RI detector model L-2490). Arabinose, fructose, fucose, galactose, glucose, glucosamine, glucuronic acid, mannose, manuronic acid, rhamnose, and xylose (Sigma-Aldrich, St. Louis, MO, USA) were used as references.

#### 4.2.2. Fourier Transformed Infrared (FTIR) Spectroscopy Analysis

For FTIR characterization, samples were pressed with KBr to form pellets in the ratio of 10% sample and 90% KBr. Pellets were analyzed by the Shimadzu FTIR-8400S spectrometer in the wavelength range 400–4000 cm^−1^ as described earlier [36].

### 4.3. Cultivation and Differentiation of 3T3-L1 Cells

The maintenance medium for 3T3-L1 cultivation was Dulbecco’s Modified Eagle’s Medium (DMEM; Gibco, Thermo Fisher Scientific, Waltham, MA, USA), containing 10% fetal bovine serum (FBS; Gibco) and antibiotics (25 μg mL^−1^ streptomycin and 10,000 U mL^−1^ penicillin, Gibco), and cells were maintained in the humidified atmosphere at 5% CO_2_ and 37 °C. The maintenance medium was replaced every 2–3 days.

For differentiation assays, cells at 80% confluency were incubated in the adipogenic medium [maintenance medium containing 1 μM dexamethasone, 0.5 mM 3-isobutyl-1- methylxanthine (IBMX), and 10 μg mL^−1^ insulin] with or without the SPS samples. After 3 days of differentiation, this medium was replaced by a maintenance medium supplemented with 10 μg mL^−1^ insulin. This medium was replaced once more because the cells were in culture for 8 days from the start of differentiation. Cells treated as described (without exposure to the SPS samples) were considered differentiated cells, and cells treated with the maintenance medium (without insulin) were considered non-differentiated cells.

For adipogenic activity screening, butanone fractions (F0.5, F0.9, and F1.8) and all fractions (CS0.1 to CS0.9) were used at two concentrations (100 and 200 µg mL^−1^). The sample that showed biological activity (CS0.2) was tested using four concentrations (25, 50, 10,0, and 200 µg mL^−1^).

### 4.4. Cytotoxicity

The effect of SPS-containing samples on 3T3-L1 toxicity was measured by the tetrazolium 3-(4,5-dimethylthiazol-2yl)-2,5-diphenyl bromide test (MTT, Invitrogen Molecular Probes, Eugene, OR, USA), according to a previous protocol [25]. Briefly, the 3T3-L1 (3 × 10^3^ cells well^−1^) were incubated in the maintenance medium with the SPS-containing samples. MTT (1 mg mL^−1^) was added after 1, 2, and 3 days of treatment. After 4 h of incubation, formazan crystals were solubilized with dimethyl sulfoxide (DMSO), and the optical absorbance was measured (570 nm) through a microplate reader (BioTek Instruments, Winooski, VT, USA). The results were expressed as a percentage relative to the non-differentiated.

### 4.5. Oil Red O Staining

Cells at 2 × 10^4^ cells well^−1^ (24 well plate) were treated with the SPS-containing sample as described in the “Cultivation and differentiation of 3T3-L1 cells” section. Then, cells were washed with phosphate buffer saline (PBS), fixed using 3.7% methanol (Sigma-Aldrich), washed twice in water, and finally treated with 60% isopropanol for cell permeabilization. Cells were stained with Oil Red O solution (0.2%; Sigma-Aldrich) and washed with water to remove the excess dye [25].

For quantitative analysis, pure isopropanol was used to elute the dye from the cells, and quantification was made by measuring the absorbance at 492 nm through a microplate reader. The results were normalized by the total cellular protein (BCA kit; Thermo Fisher Scientific, Waltham, MA, EUA), and the results were expressed as a percentage of the differentiated cells.

### 4.6. Analysis of Gene Expression

The real-time polymerase chain reaction (PCR) was employed to analyze the expression of adipogenic and lipogenic markers, as previously described [25].

After cell treatment with the SPS-containing sample (at 50 μg mL^−1^) as described in the “Cultivation and differentiation of 3T3-L1 cells” section, the total RNA was extracted, the cDNA was synthesized and used as a template in quantitative real-time PCR, using the sequences of the primers shown in Table 3, as described elsewhere [26]. The β-actin was used as endogenous gene reference, and the results were analyzed and expressed as Log2-fold change concerning the differentiated cells [37].

### 4.7. Statistical Analyses

The experiments were conducted as biological and technical triplicates. For simple and multiple comparisons, analysis of variance (ANOVA) followed by Dunnett’s test was employed using the Graphpad Prism (version 6.01). *p* < 0.05 was considered a statistically significant difference.

## Figures and Tables

**Figure 1 marinedrugs-20-00470-f001:**
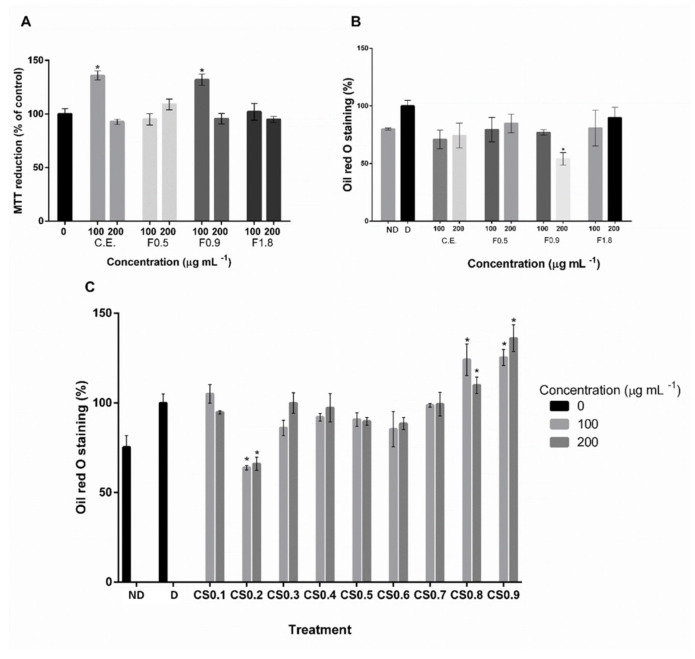
Screening of activity of SPS-containing samples from *C. sertularioides* on 3T3-L1 cells. (**A**) MTT reduction by 3T3-L1 cells treated with 100 and 200 μg mL^−1^ of crude extract, F0.5, F0.9, and F1.8 fractions along time. (**B**) Effect of SPS-containing crude extract and F0.5, F0.9, and F1.8 fractions (100 and 200 μg mL^−1^) on fat accumulation, analyzed by Oil Red O staining. (**C**) Effect of SPS-containing fractions obtained from F0.9, by ion-exchange chromatography, at 100 and 200 μg mL^−1^, on fat accumulation analyzed by Oil Red O staining. Data represent the mean ± standard deviation. * *p* < 0.05 means the result is statistically significant in relation to control. ND = non-differentiated cells; D = differentiated cells. C.E. = crude extract; F and CS = indicate the fraction according to butanone volume and sodium chloride molarity used in its preparation.

**Figure 2 marinedrugs-20-00470-f002:**
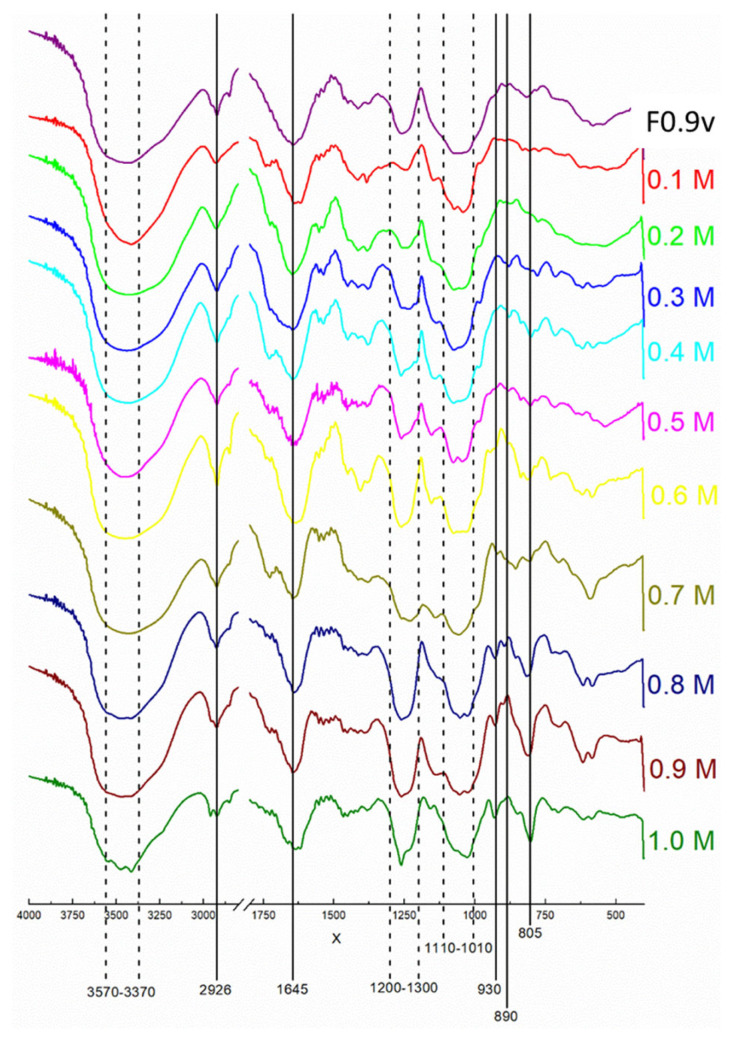
Infrared spectrum of SPS extracts from *C. sertularioides* fractions. Each color line represents different fractions obtained from the F0.9 sample by ion-exchange chromatography. The X- and Y-axis represent the wavelengths (cm^−1^) and absorbance, respectively.

**Figure 3 marinedrugs-20-00470-f003:**
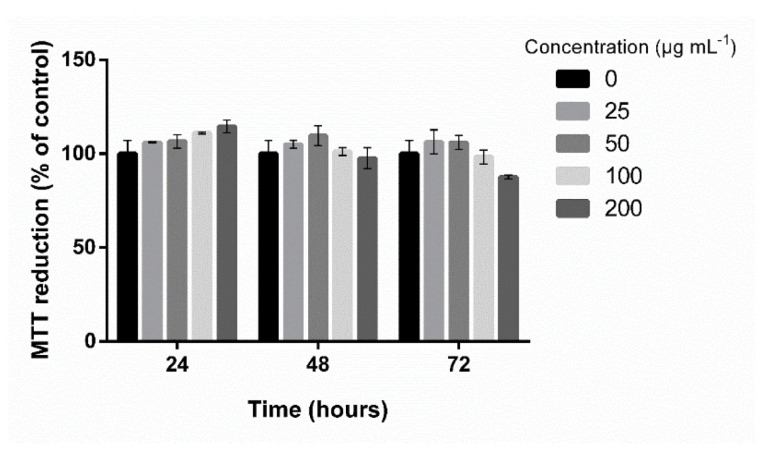
Cytotoxicity of CS0.2 fraction from *C. sertularioides* on 3T3-L1 cells. MTT reduction by 3T3-L1 cells treated with 25, 50, 100, and 200 μg mL^−1^ for 24, 48, and 72 h. Data represent the mean ± standard deviation (column bars).

**Figure 4 marinedrugs-20-00470-f004:**
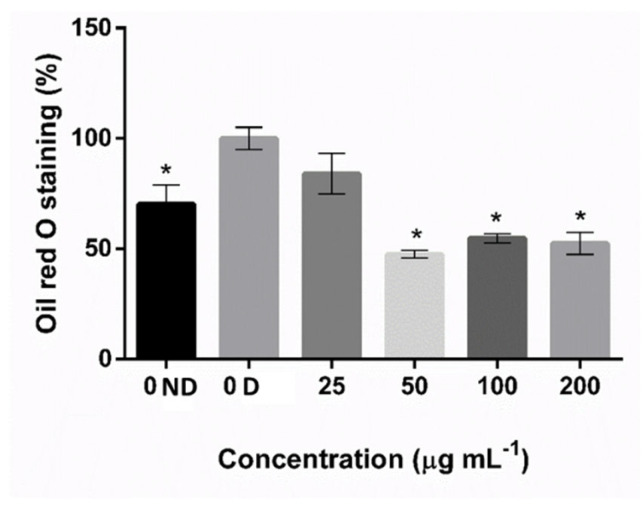
Effect of CS0.2 fraction from *C. sertularioides* on fat accumulation in 3T3-L1 cells. Fat accumulation was qualified by Oil Red O staining at 490 nm. Data represent the mean ± standard deviation. * *p* < 0.05 means the result is statistically significant in relation to differentiated cells. D = Differentiated cells; and ND = Non-differentiated cells.

**Figure 5 marinedrugs-20-00470-f005:**
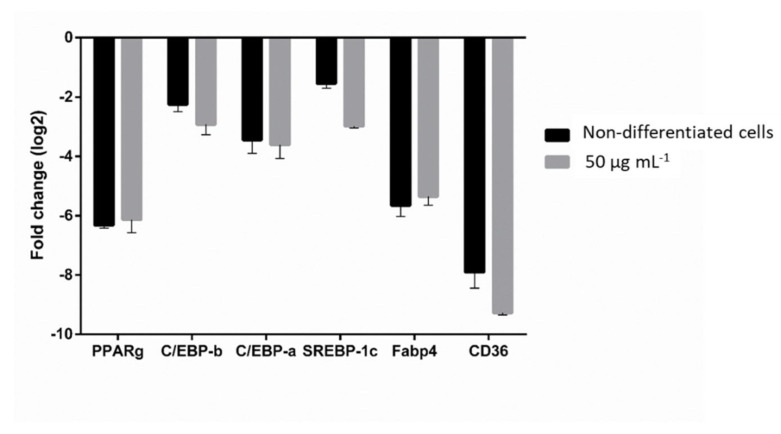
The effect of CS0.2 fraction of *C. sertularioides* on adipogenic and lipogenic markers in 3T3-L1 cells. The expression of adipogenic and lipogenic markers of cells treated with 50 μg mL^−1^ of CS0.2 fraction was evaluated by RT-PCR. Data represent the mean ± standard deviation. All genes tested were significantly downregulated in relation to the differentiated cells. Non-differentiated cells were used to validate the results.

**Figure 6 marinedrugs-20-00470-f006:**
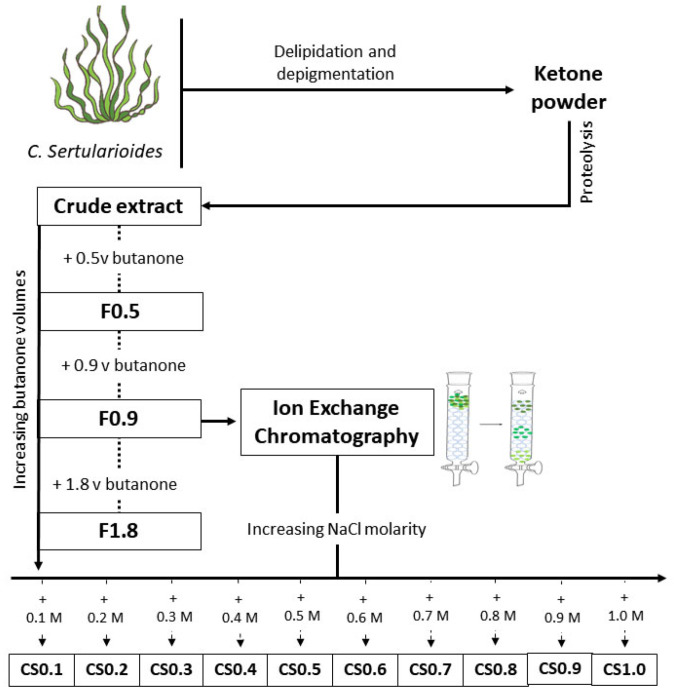
Flowchart for obtaining SPS from *C. sertularioides*. F = indicate the fraction; CS = indicate the fraction according to NaCl molarity used; v = volumes; M = molarity. NaCl = sodium chloride.

**Table 1 marinedrugs-20-00470-t001:** Chemical composition of SPS-containing extracts from *C. sertularioides* fractions obtained by precipitation with different volumes of butanone.

Extracts	Yield (%)	Total Sugars (%)	Sulfate (%)	Protein (%)	Phenolic Compounds (%)
Crude extract	- *	82.3 ± 0.9	25.5 ± 0.6	N.D	<0.1
F0.5	51.2	72.7 ± 0.3	37.2 ± 0.2	N.D	<0.1
F0.9	20.8	59.1 ± 2.2	11.2 ± 0.4	N.D	<0.1
F1.8	27.9	26.5 ± 0.2	20.5 ± 0.6	N.D	<0.1

Data represent the mean ± standard deviation. F = indicates fraction according to butanone volume used. * Represents 768.8 mg (100%) of crude extract obtained from 60 g of delipded and dried *C. sertularioides*.

**Table 2 marinedrugs-20-00470-t002:** Chemical composition of SPS-containing fractions from *C. sertularioides* obtained from F0.9 by ion-exchange chromatography.

Samples	Yield (%)	Total Sugar (%)	Sulfate (%)	Phenolic Compounds (%)	Monosaccharide Composition (Molar Ratio)					
					Glc	Rha	Fuc	Xyl	Man	Gal
CS0.1	14.7	55.2 ± 1.9	0.5 ± 0.1	<0.1	N.D.	N.D.	N.D.	N.D.	N.D.	N.D.
CS0.2	20.6	63.6 ± 1.9	2.8 ± 0.4	<0.1	1	-	-	-		-
CS0.3	12.4	58.7 ± 0.9	9.8 ± 0.6	<0.1	1	12	-	2.0		6
CS0.4	12.7	62.4 ± 2.0	12.8 ± 0.1	<0.1	1	-	-	2.0		-
CS0.5	10.7	55.2 ± 3.1	13.5 ± 0.1	<0.1	1	0.5	-	1.2	5.7	-
CS0.6	8.9	64.5 ± 1.5	15.7 ± 1.6	<0.1	1	-	-	0.1	2.2	-
CS0.7	7.9	68.0 ± 3.5	21.6 ± 1.0	<0.1	1	-	-	0.2	0.6	-
CS0.8	7.4	48.8 ± 2.3	21.6 ± 1.0	<0.1	1	-	-	2.0	2.6	3.8
CS0.9	3.0	42.7 ± 1.0	26.3 ± 1.1	<0.1	1	-	-	6.8	0.6	0.3
CS1.0	1.8	60.4 ± 1.6	17.7 ± 0.4	<0.1	N.D.	N.D.	N.D.	N.D.	N.D.	N.D.

Data represent the mean ± standard deviation. CS = reaction according to sodium chloride used; Glc = Glucose; Rha = Ramnose; Fuc = Fucose; Xyl = Xylose; Man = Mannose; Gal = Galactose. N.D. = Not determined; - = Not detected.

**Table 3 marinedrugs-20-00470-t003:** Primers used for qPCR analyses.

Gene Symbol	Primer Sequence (5′-3′)
β-actin	F: TGTCCACCTTCCAGCAGATGT
	R: AGCTCAGTAACAGTCCGCCTAG
PPARγ	F: TGCTGTTATGGGTGAAACTCTG
	R: CTGTGTCAACCATGGTAATTTCT
C/EBPβ	F: ATCGACTTCAGCCCCTACCT
	R: TAGTCGTCGGCGAAGAGG
C/EBPα	F: AGCTGCCTGAGAGCTCCTT
	R: GACCCGAAACCATCCTCTG
SREBP-1c	F: TCAAGCAGGAGAACCTGACC
	R: TCATGCCCTCCATAGACACA
Fabp4	F: CAGCCTTTCTCACCTGGAAGA
	R: TTGTGGCAAAGCCCACTC
CD36	F: GGCCAAGCTATTGCGACAT
	R: CAGATCCGAACACAGCGTAGA

F = Forward; R = Reverse.

## Data Availability

Not applicable.

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
