# Peer review of "Sulfated Glucan from the Green Seaweed Caulerpa sertularioides Inhibits Adipogenesis through Suppression of Adipogenic and Lipogenic Key Factors"

_marinedrugs, 2022, doi:10.3390/md20080470_

Round 1

Reviewer 1 Report

I recommend you to check the UCP-1, transcription factor related to the thermogenesis and lipogenesis. If it is possible, you can conclude more clearly the anti-adipogenesis effect of sulfated glucan from Caulerpa sertularioides.

Reviewer 2 Report

1.      Please keep the official short name for polysaccharide – PS. In case this Article sulfated PS as SPS.

2.      Abstract: remove details such as concentration values

3.      Results: The more details about amount of crude extract which was used to separation is needed. Also the yield of each fraction enriches this section

4.      Please use a shorter name of seaweed when the full name was used during first mention: Caulerpa sertularioides and then C. sertularioides.

5.      What is consists of fraction F0.9 except sugar (59.1%) and sulfate (11.2%) – 29.7% is needed to explanation

6.      ....F0.9 fraction was separated to 10 fractions....(there are fractions not subfractions)

7.      Lines 116-119. Please explain these sentences....”CS01 did not have its monosaccharide....” It is really unclear.

8.      Glucose – Glc (official short), Rhamnose – Rha – Mannose – Man -please use these shorts

9.      Line 149- please remove from the text

10.  Line 153. Keep the space between next line

11.  Discussion should be rewritten – please remove all details presented in Results section.

12.  Methods: it is a section 4 (check the numbering)
